# PREVIDE: A Qualitative Study to Develop a Decision-Making Framework (PREVention decIDE) for Noncommunicable Disease Prevention in Healthcare Organisations

**DOI:** 10.3390/ijerph192215285

**Published:** 2022-11-18

**Authors:** Oliver J. Canfell, Kamila Davidson, Clair Sullivan, Elizabeth E. Eakin, Andrew Burton-Jones

**Affiliations:** 1UQ Business School, Faculty of Business, Economics and Law, The University of Queensland, St. Lucia, QLD 4072, Australia; 2Centre for Health Services Research, Faculty of Medicine, The University of Queensland, St. Lucia, QLD 4072, Australia; 3Queensland Digital Health Centre, Faculty of Medicine, The University of Queensland, Herston, QLD 4006, Australia; 4Digital Health Cooperative Research Centre, Australian Government, Sydney, NSW 2000, Australia; 5Metro North Hospital and Health Service, Department of Health, Queensland Government, Herston, QLD 4072, Australia; 6School of Public Health, Faculty of Medicine, The University of Queensland, Herston, QLD 4072, Australia

**Keywords:** decision-making, preventive medicine, public health, precision public health, public health informatics, health policy, noncommunicable diseases, obesity

## Abstract

Noncommunicable diseases (NCDs), including obesity, remain a significant global public health challenge. Prevention and public health innovation are needed to effectively address NCDs; however, understanding of how healthcare organisations make prevention decisions is immature. This study aimed to (1) explore how healthcare organisations make decisions for NCD prevention in Queensland, Australia (2) develop a contemporary decision-making framework to guide NCD prevention in healthcare organisations. Cross-sectional and qualitative design, comprising individual semi-structured interviews. Participants (*n* = 14) were recruited from two organisations: the state public health care system (CareQ) and health promotion/disease prevention agency (PrevQ). Participants held executive, director/manager or project/clinical lead roles. Data were analysed in two phases (1) automated content analysis using machine learning (Leximancer v4.5) (2) researcher-led interpretation of the text analytics. Final themes were consolidated into a proposed decision-making framework *(PREVIDE, PREvention decIDE)* for NCD prevention in healthcare organisations. Decision-making was driven by four themes: Data, Evidence, Ethics and Health, i.e., data, its quality and the story it tells; traditional and non-traditional sources of evidence; ethical grounding in fairness and equity; and long-term value generated across multiple determinants of health. The strength of evidence was directly proportional to confidence in the ethics of a decision. *PREVIDE* can be adapted by public health practitioners and policymakers to guide real-world policy, practice and investment decisions for obesity prevention and with further validation, other NCDs and priority settings (e.g., healthcare).

## 1. Introduction

Non-communicable diseases (NCDs)remain the largest global public health threat. People of all ages and countries are affected by NCDs; they contribute to 71% of deaths globally [1]. Most of this burden is caused by four major NCDs: cardiovascular diseases, cancers, chronic respiratory diseases, and diabetes [2]. Overweight and obesity is a significant and common risk factor to all four major NCD yet has now also emerged under global scientific consensus as a chronic, relapsing, and progressive disease process [3]. In Australia, obesity is the second highest contributor (next to tobacco use) to the preventable burden of NCDs [4]. Obesity is caused by complex, interwoven interactions between social, environmental, behavioural, biomedical and commercial determinants of health [5]. Treatment is difficult and often unsuccessful, and in a public health context, prevention has now been prioritised nationally in Australia with the release of a National Obesity Strategy (March 2022) [5].

In the context of obesity prevention, traditional models of public health have relied upon evidence generated from crude retrospective data captured in in point-prevalence surveys, retrospective cohorts, clinical trials or disease-specific research databases or registries [6,7]. Such cross-sectional data may include geographically measured prevalence of obesity, fruit and vegetable intake or physical activity levels—typically individual risk factors. The rapid evolution of obesity prevalence and knowledge in recent decades—and its manifestation as a relapsing and progressive disease process with complex, multisectoral origins largely outside individual behaviours [3]—is incompatible with decision models based upon historical data, and risks the possibility of redundant and ineffective decisions.

Public health innovation is needed to effectively address obesity. New sources of real-world data (e.g., electronic medical records, wearables) and digital technologies are the foundation for innovative ‘digital public health’ [8,9] approaches that can augment traditional policy and practice with new levels of precision [10]. Digital public health is an opportunity to reinvigorate traditional public health towards *precision* public health approaches that target communities based upon risk and need, and in real-time, i.e., *the right intervention for the right population at the right time* [10]. Digital public health has revolutionised communicable disease response and management (e.g., COVID-19) [11,12,13] but benefits have not been realised for NCDs (e.g., obesity, cardiovascular disease, type 2 diabetes) where the health burden—and thus opportunity—is highest. International research has begun to advance this significant but currently unmet opportunity by (a) justifying digital health as a strong enabler to achieving precision public health [7] (b) developing a strategic roadmap for precision public health of NCDs [10] (c) identifying and mapping useable data assets for precision public health of obesity [6].

New population health analytics tools and strategic visions to advance precision public health for NCDs have been developed [10]; however, an empirical gap remains in understanding *how* their end-users—healthcare organisations—make decisions to support contemporary policy and practice for NCD prevention. Addressing this gap will mean digital public health can be tailored to organisational decision workflows and ultimately enhance the translation and impact of innovative tools, products and precision models of care. In designing our research question and aims to address this gap, we referred to the health improvement decision-making triangle, a seminal theory (2008) of decision-making for public health [14]. An assumption is that this existing theoretical framework will provide theoretical grounding for understanding how NCD prevention decisions are made but we expect that—due to empirical advancements in public health theory and practice—a new, contemporary framework that reflects these advancements may be uncovered, and thus justifies our exploratory approach.

Our research question is *“how do healthcare organisations make decisions for NCD prevention?”.* To situate our study, we focus on the context of Queensland, Australia. Queensland offers a representative site because obesity rates remain troubling and there is significant political and organisational inertia to address the problem. We hypothesised that preventive decision-making comprises a balance between evidence and community need.

Our aims were:To (a) explore and (b) compare decision-making in two healthcare organisations (clinical and public health) for NCD prevention using a qualitative approach and Queensland, Australia as the use case setting.To develop a contemporary decision-making framework for NCD prevention in healthcare organisations.

## 2. Materials and Methods

### 2.1. Design

This study was cross-sectional and qualitative (phenomenological) in design and comprised individual semi-structured interviews to investigate its research question. A qualitative design was chosen to derive rich perspectives and experiences of decision-making for NCD prevention from participants in healthcare organisations. A phenomenological method (i.e., interviews) suited exploration of decision-making—a fluid and dynamic real-world process—and allowed deep analysis of the NCD prevention and unique organisational contexts [15]. We adhered to the Consolidated criteria for Reporting Qualitative research (COREQ) checklist (Appendix A) [16].

### 2.2. Participants and Recruitment

Participants were recruited from the state public health service (CareQ) and health promotion organisation (PrevQ). CareQ is responsible for clinical healthcare delivery. PrevQ is a recently established (July 2019) statutory health promotion agency dedicated to obesity prevention. Participants were change-agents: actors whose leadership position influences decision-making, strategy and policy at an organisational level. Participants were targeted across three groups according to seniority (1) Executive (2) Director/Manager (3) Project/Clinical Lead. Executive participants were members of executive leadership and provided governance, strategic and directional leadership to the organisation. Director/Manager participants were members of senior leadership and provided team-based strategy and direction to digital healthcare delivery (CareQ) or health promotion programs and policies (PrevQ). Project/Clinical lead participants were frontline leaders and conducted project-specific activities in small teams.

Participants were first approached at an organisational level. We employed a purposive sampling strategy due to the specificity of our target population. A snowballing approach was also used to identify other potential participants beyond those initially approached. As per contemporary recommendations for qualitative research, we did not adhere to sample size numerical guidelines or norms [17]. Our final sample size (*n* = 14) was contextual; it represented the small number of eligible and consenting participants in high-level, statewide organisational roles in CareQ and PrevQ.. The study team delivered a short presentation to eligible change-agents at CareQ and PrevQ on the study rationale, aim, design, risks, benefits and significance. Potential participants could then express interest via email or phone. A detailed explanation of the study purpose, methods, demands, risks and outcomes was then provided. Participants provided informed consent online and completed a short demographics questionnaire. Participants who expressed initial interest but did not engage further were followed up with two emails. No interviewer-participant relationship was established prior to data collection. No rewards or incentives were offered to participants.

### 2.3. Data Collection

Demographic questionnaires were completed by participants via a secure, online survey tool—‘Checkbox’. Demographic data included: role category (Executive, Director/Manager, Project/Clinical Lead), professional background (clinical, public health, information technology, research, project, operational), length of time in current role (0–6 months, 6–12 months, 1–2 years, 2–5 years, >5 years) and years of total experience (<2 years, 2–5 years, 6–10 years, 11–20 years, 21–30 years, >30 years).

Videoconference semi-structured interviews were conducted in a closed room at a university research centre by two digital health and obesity researchers (KD, PhD, Research Assistant and OJC, PhD, Research Fellow). Interviews were audio recorded and transcribed verbatim by an external transcription service. Interview duration was 22–37 min. Transcribed interviews were not returned to participants.

### 2.4. Semi-Structured Interviews

Semi-structured interview design (Table 1) was guided by the health improvement decision-making triangle [14]. Questions were separated into three categories: ethics, evidence, theory (i.e., experiential and professional judgment). Lead and probing open-ended questions related to the research question were devised by the research team. Prior to commencing interviews, the questions were piloted by one researcher (KD) with three PhD candidates in digital health. Before commencing each interview, clarification was provided to participants that in the context of this study, preventive decision-making refers to: CareQ—*“healthcare delivery decisions that target disease prevention at all levels”*; PrevQ—*“health promotion decisions that seek to improve healthy weight, food and drink choices”*. Decisions were defined at the organisational level, and may have related to investment, intervention design, choosing target settings or conducting research studies, as examples. In the participant feedback interviews, participants were invited to a semi-structured discussion about their overall impressions of the results, and perceived strengths, weaknesses and gaps.

### 2.5. Data Analysis

Interview data were analysed in a two-stage approach.

#### 2.5.1. Stage One—Unsupervised Machine Learning

The first stage of analysis was undertaken via the text analytics tool ‘Leximancer’ (v4.5) [18]. Leximancer applies an unsupervised machine learning algorithm that uncovers networks or patterns of word- and name-like terms in a body of text [19]. It then generates interconnections, structures and patterns between terms to develop ‘concepts’—collections of words that are linked together within the text—and group them into ‘themes’—concepts that are highly connected [19]. Leximancer displays the inter-relationships between concepts and themes visually. Its advantages include expedition of the early stages of qualitative analysis and enabling researchers to obtain a representation of the data that is not biased by their own preconceptions.

One researcher loaded the observational data into Leximancer and created an initial concept map (a birds-eye analysis of the text) without altering any settings. Leximancer’s tagging functionality was also used to stratify and analyse data according to organisation and position seniority as subgroups. Concepts were then iteratively reviewed for ‘lexical value’ and removed where appropriate. Concepts were merged where relevant (e.g., ‘understand’ and ‘understanding’). The concept map was initially observed at a summary level through ‘zooming out’ then individual themes and concepts were investigated in more detail by ‘zooming in’ as described by Haynes et al. [20]. Multiple theme sizes were trialled to arrive at the final concept map where a theme size of 60% was used.

#### 2.5.2. Stage Two—Researcher-Led Interpretation

A second stage of researcher-led interpretive analysis was applied to the preliminary themes and concepts identified by Leximancer’s text analytics. Leximancer was queried for samples of text that supported each preliminary theme. Relevant text was extracted by one researcher. Manual inductive, thematic analysis was performed by independently by two researchers; results were then discussed in two iterative rounds of grouping to generate new, interpretive themes and sub-themes based on contextual understanding of the field (chronic disease prevention). Researchers performed a ‘cluster and name’ process whereby themes and sub-themes were synthesised into a preliminary decision-making framework for NCD prevention in healthcare organisations ready for feedback.

#### 2.5.3. Developing the Decision-Making Framework

The preliminary framework was refined in two iterative phases: (1) presentation, discussion and refinement with the research team (2) individual consultation with select executive participants (*n* = 3) from CareQ and PrevQ. Research team and participants feedback was synthesised into a final proposed decision-making framework for NCD prevention in healthcare organisations.

## 3. Results

### 3.1. Participant Characteristics

A total of 18 organisational change-agents expressed interest to participate in the study. Of these, 14 change-agents consented and participated in the interview (Table 2). The primary reason for non-response was lack of response to follow-up contact. Most change-agents represented health promotion and public health in PrevQ (10, 71%) versus state healthcare delivery in CareQ (4, 29%).

Change-agents mostly held roles as Project/Clinical Lead (7, 50%), then Director/Manager (5, 36%) and Executive (2, 14%). Public health (5, 36%) and clinical (4, 29%) were the most commonly reported professional backgrounds. Most change-agents had spent 1–2 years (6, 43%) or greater (6, 43%) in their current role. Change-agents were highly experienced with most reporting 11–30 years of total experience (10, 71%).

### 3.2. Stage One—Identifying Preliminary Themes and Concepts Using Leximancer

Figure 1 shows the inter-topic concept map derived from the interview data. From Leximancer’s machine learning analysis, this map illustrates themes as coloured bubbles that are heat-mapped according to their frequency (‘importance’), with warmer colours (e.g., red, yellow) indicating higher importance and cooler colours (e.g., blue, purple) lower importance. Concepts are displayed as dots within each coloured theme bubble and inter-linked across themes. Closer proximity of the coloured bubbles or concept dots indicates higher co-occurrence.

At a theme size of 60%, six themes were automatically derived (in order of identified ‘importance’): data, evidence, health, ethical, investment, access. The analysis identified 44 total concepts within all themes. The top 10 most frequent concepts were: data, evidence, people, work, health, need, use, example, understand, look.

### 3.3. Stage Two—Final Themes and Sub-Themes of Preventive Decision-Making in Healthcare Organisations

A total of four themes (Data, Evidence, Ethics, Health), 11 sub-themes and one minor theme (Evidence-Ethical) were manually derived from stage two (researcher-led interpretation of the Leximancer outputs) data analysis (Table 3).

Figure 2 presents a conceptual decision-making framework for NCD prevention in healthcare organisations *(PREVIDE—PREVention decIDE)* based on all themes and iteratively refined with the research team and change-agents. ‘Investment’ (a theme derived from stage one) was interpreted as the final decision arising from a complex interrelationship between Data, Evidence, Health and Ethics themes, rather than a true theme. Ultimately, decision-making results in either (a) investment of time, resources, money and/or organisational inertia (b) no action, i.e., neutral position or (c) disinvestment, which is a withdrawal of investment.

#### 3.3.1. Theme 1—Data

Theme 1 and its sub-themes were derived from the strongest ‘Data’ theme identified by Leximancer (see Box 1 for supporting participant quotes). Overall, data is a powerful tool used by change-agents to make trusted decisions and tell meaningful stories to promote prevention activities. Storytelling was essential to prevention and included stories of success and impact, describing a disease problem and community voices. Gaps in data were consistently identified, including data availability, particularly for data that exists in communities (i.e., social, environmental, behavioural) where there has been little historical investment in digital infrastructure and data quality (e.g., greater geographical coverage). Gaps were identified in communities and areas where the need is strongest, i.e., for priority, underserved populations. Higher quality data that addressed these needs was seen to generate higher confidence and precision in decision-making.

Box 1Participant quotes supporting theme 1 (Data).
*“It’s using the best advice, the best evidence, the best…
pulling all the data, everything that you’ve got to make the most…
well-informed choice or recommendation or decision at that particular point
in time”*

*“If it’s out by a few percent because of some limitations
in that data collection that’s not necessarily a big deal, it’s still telling
a story that’s an important story”*

*“If you wanted to make a decision on anything, there’s
literally so much data out there… but not all of it is usable”*

*“If we had that sort of data we could be a lot more
confident in our decision-making around strategies and it would help use be
more preceise in what we would invest in”*


#### 3.3.2. Theme 2—Evidence

Theme 2 and its sub-themes were derived from the ‘Evidence’ theme identified by Leximancer (see Box 2 for supporting participant quotes). Change-agents described evidence as critical to decision-making, especially as a first step to a designing an innovation or new intervention. Inductive analysis revealed two types of evidence (1) traditional—informed by academic literature, research and government or university reports; and (2) non-traditional, informed by ‘innovation factor’, alignment to organisational strategy, experience (professional and community), assumptions, and theory. Traditional evidence was the ‘holy grail’; however, evidence for prevention (i.e., predict-prevent) was highlighted as scarce in comparison to the amount—and strength—of evidence to support treatment decisions (i.e., break-fix), and so decisions required an equitable balance between traditional and non-traditional sources.

Box 2Participant quotes supporting theme 2 (Evidence).
*“That was certainly the evidence that sealed the deal on
the decision we made at project committee, absolutely”*

*“So we looked at the evidence and decided as an
organisation, a prevention lens, or an investment in prevention… this
programme would lead to—likely lead to some better outcomes for them in the
long-term*

*“If you want to be innovative you can’t necessarily say
‘every decision I’m going to make has to be evidence-based’”*

*“Do you say ‘do we wait until we have randomised
controlled trials?’ or do you say ‘I think on balance of evidence, this would
be a prudent course of action and is unlikely to be harmful’, you know”*

*“Obviously there’s empirical evidence which is really
important to draw on but then there’s also understanding what other pieces of
evidence exists… evidence in terms of community experience”*


#### 3.3.3. Minor Theme—Evidence-Ethical

A minor theme was derived from stage two analysis that demonstrated synergy between Evidence and Ethics themes (see Box 3 for supporting participant quotes). The strength of all evidence, both traditional and non-traditional, was interpreted as directly proportional to confidence in the ethics of a decision. An ethical decision has a foundation of evidence; gathering appropriate information with all relevant stakeholders and perspectives considered, especially from those who are affected by the decision.

Box 3Participant quotes supporting minor theme (Evidence-Ethics).
*“To me, in terms of having ethical decision-making, that
it’s ethical decision-making would also be something that is based on
evidence and isn’t just subjective”*

*“Ethical decision-making to me would be ensuring that I
have all the information I need from them… so a deep analysis around
strategic alignment, risk, value proposition for us, applying the investment
frameworks we’ve developed so there’s criteria around that”*


#### 3.3.4. Theme 3—Ethics

Theme 3 and its sub-themes were derived from the ‘Ethics’ theme identified by Leximancer (see Box 4 for supporting participant quotes). Ethical decisions were ultimately *fair* and underpinned by a fundamental question—*“is it the right thing to do?”*. Fairness was linked to prioritizing communities with an equity lens, i.e., those with disproportionately difficult social, environmental and disease circumstance. Change-agents expressed a sense of responsibility—to the taxpayer, community, and their organisation—that guided their decision-making. Ethics were discussed at a (a) personal level, where variation between individuals was possible (b) community level, where community *need* and equity was absolute, and (c) organisational level, where ethics were principled, such as ‘do no harm’ and ‘the right choices to benefit the most people’, and based around organisational structures, such as strategy, investment frameworks and policy.

Box 4Participant quotes supporting theme 3 (Ethics).
*“That’s an ethical decision because we know it’s
strategically aligned, it’s using the right governance and it’s a responsible
decision. Is it the right thing to do?”*

*“I guess the bottom line is we are almost wholly and
solely funded by government, that is we’re using public funds, taxpayer’s
money*

*“Not just what I think is ethical, but certainly a whole
lot more about the consumers or the participan… that they are aware and that
it’s ethical from their perspective tacking into account their cultural
diversity, everything else*

*“From a definitional point of view, ethical decision-making
is about making the right choices that are going to benefit the most people”*

*“Ethical decision-maknig does mean looking at fairness so
one huge thing of ethics is making sure you are fair so that we do things for
the best benefit for most”*


#### 3.3.5. Theme 4—Health

Theme 4 and its sub-themes were derived from the ‘Health’ theme identified by Leximancer (see Box 5 for supporting participant quotes). Change-agents identified that understanding determinants of health (e.g., education, housing, justice), rather than traditional measures of health (e.g., body mass index), is critical for preventive decision-making. There was clear focus on understanding and targeting risk factors for poor health across multiple sectors (e.g., health, education, housing, environment, transport) and considering political context and community context before making preventive decisions. Prevention was discussed in the context of health dollars, and how efficient prevention means lower spend and a higher return on investment in health and wellbeing in the long-term.

Box 5Participant quotes supporting theme 4 (Health).
*“Often it’s been around wellbeing and children so
housing, employment, justice, those kinds of equity drivers of good health
and wellbeing”*

*“I think what’s missing, from my perspective in terms of
prevention, it’s really considering that health is not just health now, it’s
the social determinants of health linking mobility data, transport data,
activity data”*

*“One of the parts of ethics is doing things efficiently
so we only have a certain health dollar spend”*

*“We really needed to capture better our input, what
staffing was put into that, to the clinic, how much time was spent with
families—to really be able to flesh out if you invest this much, this is the
economic output because we all know health relates to dollars in the end”*


#### 3.3.6. Participant Feedback

Overall, select executive participants (*n* = 3) from CareQ and PrevQ felt the framework *(PREVIDE)* was fair, logical, and appropriately representative of the real-world context. The four pillars—Data, Evidence, Ethics, Health—were viewed as consistent across heterogenous healthcare settings, such as treatment and management services (i.e., hospitals, primary care). It was suggested that each pillar does not have equal strength—and thus contribution to—a final decision, rather, the weighting of each pillar may and likely should fluctuate and be tailored to each decision context. Considering the pillars as equal contributors to a decision was viewed as a weakness and risk.

Translation of *PREVIDE* into healthcare organisations was considered a priority action. Translation avenues were suggested as:Other decision-making settings in healthcare (e.g., point-of-care)An investment model for prevention (and an evidence-based mechanism to increase investment)A decision-making tool that can help guide investment (or disinvestment) at an organisational level, and not just a tool that describes how decisions are made.

Enablers to successful translation were using real-world use cases to ground *PREVIDE* in action and considering organisational readiness for decision workflow disruption.

### 3.4. Subgroup Analysis

#### 3.4.1. Organisation

Figure 3 presents the inter-topic concept map derived from the interview data stratified by organisation (CareQ vs. PrevQ) (see Box 6 for supporting participant quotes).

At a theme size of 56%, five themes were automatically derived (in order of identified ‘importance’): data, evidence, trying, information, population. CareQ and PrevQ were both linked to all themes. The top 5 linked concepts for CareQ were: research, unclear, looking, approach and outcomes. The top 5 linked concepts for PrevQ were: access, system, used, understand and need.

CareQ was driven by evidence-based, research-driven approaches to decision-making, and collaborated with research organisations to help build evidence and guide investment decisions. PrevQ decision-making was often discussed at a systems and population level; the need to generate prevention impacts for those most in need that can positively change the system towards a ‘predict-prevent’ model of healthcare.

Box 6Participant quotes supporting subgroup analysis (organisation).
*“I don’t believe in approaches that have no evidence-base
at all… there has to be a foundation and I go to the research literature
looking for that” (CareQ)*

*“We just didn’t have the data points that we needed to
show impact at the level that was required, at the system level…” (PrevQ)*


#### 3.4.2. Role Category

Figure 4 presents the inter-topic concept map derived from the interview data stratified by participant role category: Executive (Exe), Director/Manager (Dir), Project/Clinical Lead (PCL) (see Box 7 for supporting participant quotes).

At a theme size of 56%, five themes were automatically derived (in order of identified ‘importance’): data, people, ethical, activity, investment. Executive was linked to all themes, with the top 5 linked concepts including: framework, access, used, approach and data. Director/Manager was also linked to all themes, with the top 5 linked concepts including: research, unclear, looking, approach, outcomes. Project/Clinical Lead was linked to four themes: data, people, ethical and activity. The top 5 linked concepts were access, system, used, understand, need.

Executive change-agents grounded their decisions in structured frameworks, based upon data and evidence, that could be applied consistently to different contexts. Director/Manager’s referred to research as one pillar of decision-making among other forms of evidence, such as practise-based experience and anecdotal evidence, and were motivated by generating and measuring health outcomes. Project/Clinical Leads were focused on data outside health, such as community access to healthy environments, and access to additional high-quality data (e.g., economic, council) to support decisions where need was greatest.

Box 7Participant quotes supporting subgroup analysis (role category).
*“From the start I wanted to make sure that had evidence,
research and learning framework around everything”*

*“You’ve got the research evidence, you’ve got the
scientific evidence… and then you’ve got your practise evidence, and that
kind of goes into more andecdotal clinical evidence”*

*“I would really like to see us have more access to
economic data or the cost of things”*


## 4. Discussion

### 4.1. Main Findings

This study engaged 14 change-agents from healthcare organisations in Queensland, Australia to explore how decisions are made for NCD prevention. Overall, preventive decision-making is driven by four domains that vary in weight depending upon the decision context: data, its quality and the story it tells; traditional and non-traditional sources of evidence; ethical grounding in fairness and equity; and long-term value generated across multiple determinants of health. Whilst change-agents were ethically bound to make fair and just decisions regardless of supportive data and evidence, their level of confidence in the ethics of a decision was directly proportional to the amount of traditional and non-traditional evidence supporting the decision. The framework *(PREVIDE)* was seen to have pragmatic value by change-agents as a decision-making tool—beyond a traditional theoretical framework—to guide prevention investment (or disinvestment) by healthcare organisations, including at the individual patient and community level. *PREVIDE* is a first step in modelling organisational decision-making for NCD prevention, using obesity as a public health use case and Queensland, Australia as the use case setting.

### 4.2. Comparison to Literature

The literature on decision-making for health promotion and public health has evolved considerably over the previous two decades. Seminal theories (1997) focused solely on evidence to drive decisions about the care of communities and populations—a top-down approach [21]. Later work included data and information systems (1999 [22]) by Brownson et al. and community preferences (2004 [23]) by Kohatsu et al. as core components to evidence-based public health decision-making. When taken together, these foundational principles of evidence, data and community need remain strong pillars of decision-making in public health. These principles have matured to reflect societal transformation and, vice versa, societal transformation has likely influenced decision-making principles. Our understanding and acceptance of what constitutes evidence has evolved as digital infrastructure creates massive amounts of real-world big data [7], and propagated near-universal inertia to achieving health equity [24].

In 2009, Tannahill developed a new framework for decision-making for health promotion, public health, and health improvement: the decision-making triangle with the pillars evidence, ethics and theory [14]. Subsequently, Carter et al., published a framework tailored to health promotion that grounded decision-making in evidence, ethics and values [25]. Evidence is the foundational pillar of decision-making in both frameworks. Tannahill described evidence ‘strands’ that related to health issues, risk factors and effectiveness and base upon traditional research evidence. Our findings propose that in the context of NCD prevention, non-traditional sources of ‘evidence’ such as level of innovation, strategic alignment, experience (professional and community), assumptions, and theory are perceived as real evidence sources—they carried similar weight to traditional evidence sources (e.g., research)—and were found to significantly influence decision-making. This could explain the absence of Tannahill’s [14] ‘theory’ pillar in *PREVIDE* as theory collapsed into ‘evidence’ as a non-traditional source important for prevention decision-making. One reason for this finding lies in the prevention paradox; prevention (i.e., predict-prevent models of healthcare) is empirically more effective than break-fix models of healthcare [7,26]; yet, there is an absence of evidence for prevention and generating this evidence is difficult as measuring an outcome that never occurred is inherently challenging. This prevention paradox likely fuels dependence upon non-traditional sources of evidence to guide public health policy and practice.

Ethics as a pillar of decision-making for health promotion and public health remained consistent between previous frameworks and *PREVIDE*—our proposed framework [14,25]. Tannahill [14] promotes ethical principles, including equity, respect, empowerment, participation and openness, and social responsibility as critical prior to using either evidence or theory to inform health promotion actions. This was reflected in our study findings; change-agents described overwhelming responsibility to make decisions based upon what is *just* and *right*, underpinned by accountability to the communities they serve and regardless of data and evidence availability. Carter et al. [25] developed the ethics pillar further to argue that evidence and ethics are implicitly related; evidence-based practice is more likely ethical. Our study corroborated this finding in the context of NCD prevention; the strength of all evidence, both traditional and non-traditional, was proportional to change-agent confidence in the ethics of a decision.

One unique component of *PREVIDE* is ‘Data’ existing as an independent pillar separate to ‘Evidence’. Historically, data may have nested within the evidence domain; however, rapid digital transformation of society and healthcare is creating new models of *precision* public health based upon real-world and big data [6,10,27]. Data is not synonymous with evidence; for example, lifestyle wearable devices such as smartwatches generate personal health information in real-time but most do not meet the data accuracy regulatory standards required for medical equipment [28]. More data does not also automatically mean more information, and information is not automatically evidence. With technical skill and context, data can become information that informs a decision, but evidence generation requires consistent academic rigor over time. Change-agents in the present study described the ability of data to ‘tell a story’ about a community or health issue, and the trustworthiness of high-quality data as offering unique value to decision-making.

‘Health’ as a pillar of decision-making was another finding unique to *PREVIDE*. Change-agents were clear that understanding and targeting risk factors for poor health across multiple sectors (e.g., health, education, environment, transport) was core prevention practice for NCDs. This was not explicitly reflected in the frameworks of Tannahill [14] and Carter et al. [25], perhaps because public health evidence relating to the determinants of health has evolved significantly in the previous decade. For example, the recently launched *National Obesity Strategy (2022–2032)* [5] in Australia strongly reflects how social, environmental, and commercial determinants of health influence obesity and necessitate a coordinated systems approach to prevention as opposed to the outdated behaviour-driven model that assigns individual blame and responsibility [25]. Our understanding of health has matured beyond simple biomedical measures, such as body mass index, and contemporary prevention decision-making has matured proportionately.

### 4.3. Implications for Practice

Theoretical models of decision-making for public health and health promotion have now matured into pragmatic decision frameworks. Creating evidence-informed policies and recommendations requires translating theoretical pillars such as data, evidence, ethics and health, into real-world action. Contemporary ‘Evidence-to-Decision’ frameworks exist to assist public health practitioners and policymakers formulate evidence-driven decisions, such as in environmental health [29] and health system [30] contexts. Key criteria consistent across prominent frameworks typically assess the priority or problem severity, benefits and harms, certainty of evidence and resource implications, with fewer frameworks including equity and acceptability as core pillars [29]. The GRADE Evidence-to-Decision framework is thorough and considers health equity, intervention acceptability, feasibility and economic benefit along with traditional criteria. Equity and return on investment were sub-themes identified in our present study, with change-agents highlighting that equity is the ‘theme’ of prevention, and efficient prevention means greater long-term economic impact. Overall, there are consistencies between *PREVIDE* and existing decision frameworks (e.g., equity, evidence); however, prevention requires further consideration of non-traditional evidence, which contributes significantly to decision-making.

Change-makers in our study were quick to identify the translation potential of *PREVIDE* into healthcare organisations to embed contemporary decision-making workflows into routine prevention practice. One real-world example is use of the GRADE Evidence-to-Decision framework to strengthen decision-making in national food fortification programming in Nigeria [31]. In an Australian context, state health promotion organisations tasked with implementing Australia’s *National Obesity Strategy (2022–2032)* [5] could benefit from a decision framework optimised for prevention as the *Strategy* prioritises developing new prevention policies and interventions at a system and population level, and difficult investment decisions will need to be made at a high organisational level. Strategy and policy decisions at a healthcare organisation level can be guided by political priorities, such as leadership, budget and beneficiaries, a factor not identified by change-makers in our present study [32]. Unavoidable competing interests among healthcare actors (profit and nonprofit enterprises, government, healthcare executive) is one reason for health systems’ failure to implement ‘predict-prevent’ models of care [33]. It may be necessary to account for political inertia in decision-making frameworks for NCD prevention and provide strategies to quantify and monitor political influence that may risk decision ethics.

### 4.4. Strengths and Limitations

To our knowledge, our study is the first to develop a contemporary decision-making framework to support healthcare organisations in NCD prevention. We adopted an evidence-based qualitative approach that guided development of *PREVIDE*. Data analysis was strengthened by a machine learning (via Leximancer) approach to provide initial text analytics that informed researcher-led inductive analysis. The primary limitation of this study was the overrepresentation of change-makers from PrevQ, the state health promotion organisation, compared to CareQ, the state healthcare system. Decision-making for prevention in clinical settings may thus have been underrepresented and clinical validation of *PREVIDE* would be required to justify application at a hospital, health service or point-of-care environment.

## 5. Conclusions

This study provides empirical evidence for how healthcare organisations in Queensland, Australia—both health promotion and healthcare delivery—make decisions for NCD prevention. Overall, decisions are driven by four core pillars that vary in strength depending upon the decision context: data, its quality and the story it tells; traditional and non-traditional sources of evidence; ethical grounding in fairness and equity; and long-term value generated across multiple determinants of health. Confidence in decision ethics was directly proportional to the supporting evidence-base. As a first step in a nascent area, we developed a pragmatic framework *(PREVIDE)* that can be adapted by public health practitioners and policymakers to support policy, practice and investment decisions for obesity prevention and with further validation, broader NCDs (e.g., cardiovascular disease, type 2 diabetes) in healthcare organisations. Researchers may also use *PREVIDE* and its decision-making pillars to guide new research questions and research uptake in prevention organisations. Future research can prioritise validation of *PREVIDE* in real-world settings and other NCD contexts to support emerging precision public health models of care and evaluating its impact on decision-making confidence and impact.

## Figures and Tables

**Figure 1 ijerph-19-15285-f001:**
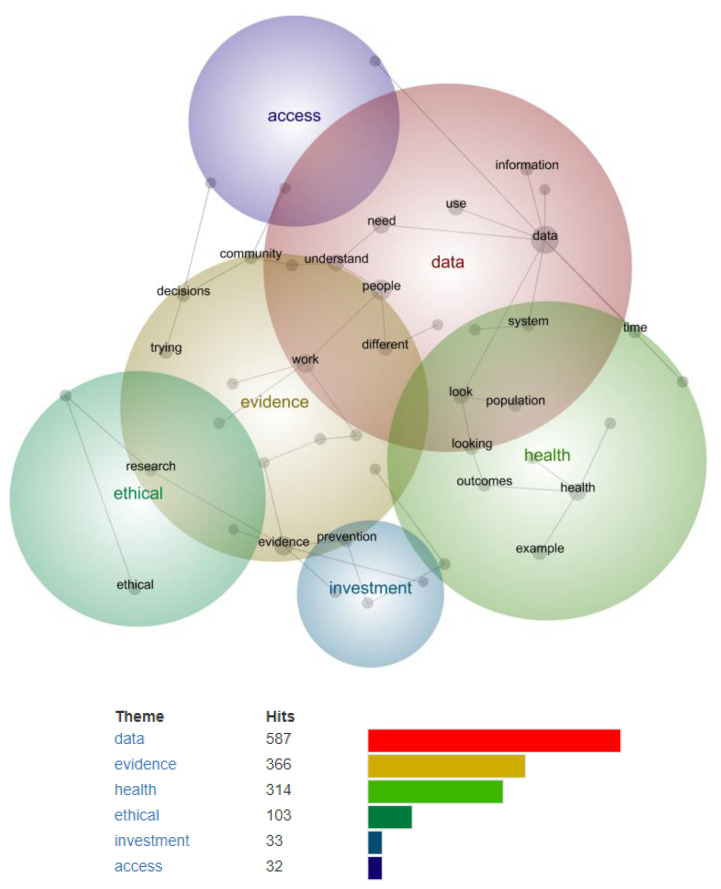
Concept map from Leximancer data analysis (stage one).

**Figure 2 ijerph-19-15285-f002:**
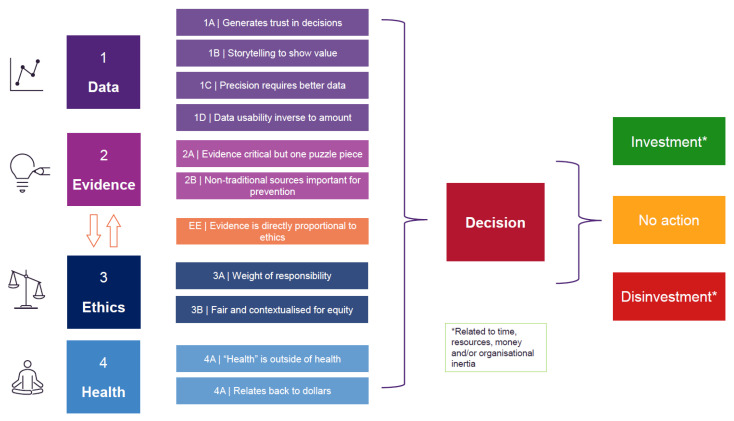
PREVIDE (PREVention decIDE)—a decision-making framework for noncommunicable disease prevention in healthcare organisations.

**Figure 3 ijerph-19-15285-f003:**
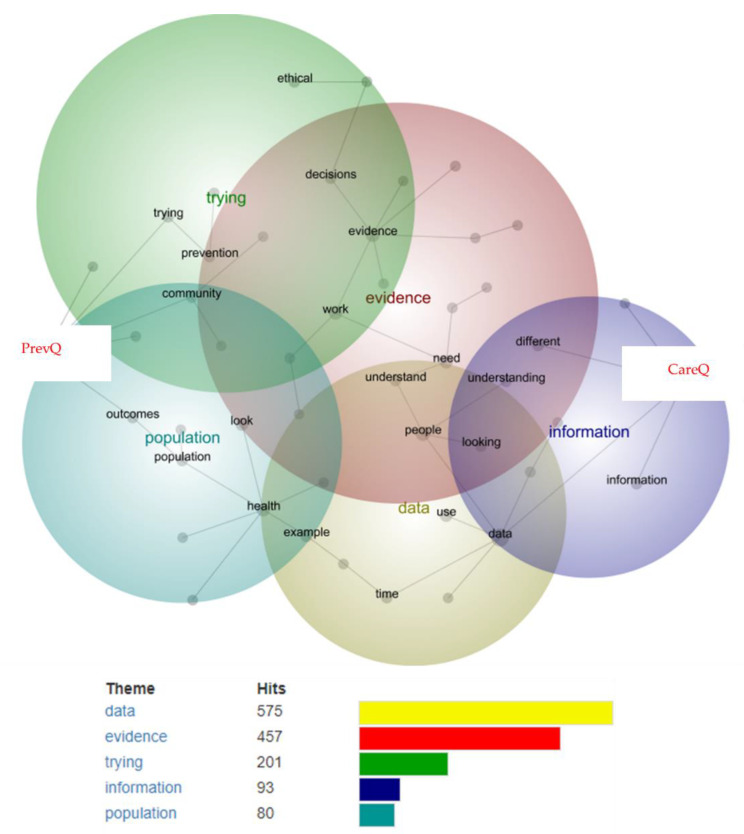
Concept map from Leximancer data analysis stratified by organisation (PrevQ, CareQ).

**Figure 4 ijerph-19-15285-f004:**
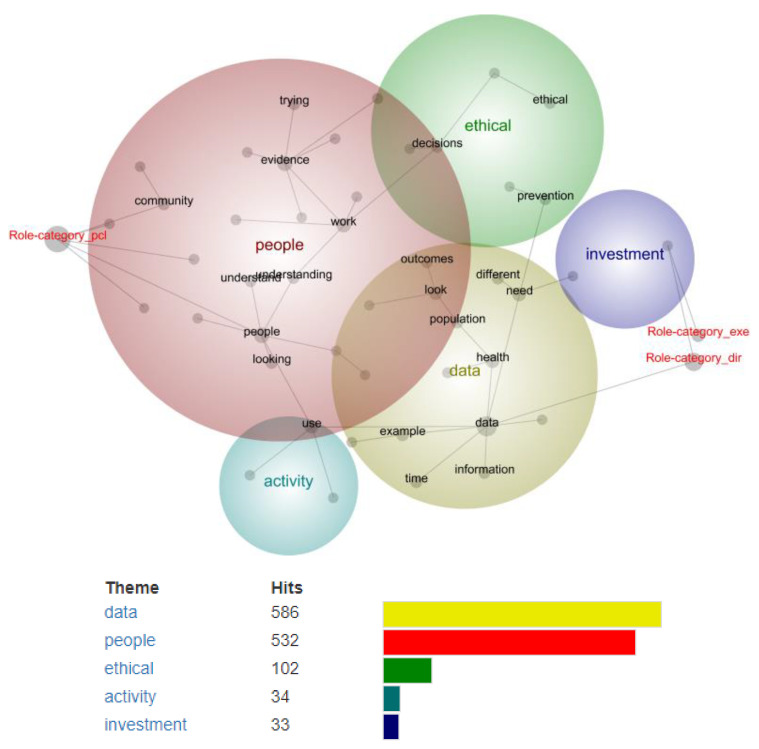
Concept map from Leximancer data analysis stratified by role category (Exe, Dir, PCL).

**Table 1 ijerph-19-15285-t001:** Semi-structured interview design.

Domain	Questions	Prompts
*“In the context of this study, preventive decision-making refers to any organisational decision that targets disease prevention at all levels of healthcare and public health.” Decisions may use a combination of ethics, evidence and theory.*		
Ethics	What is your understanding of ethical decision-making?	For prevention? In the context of your organisation?
	How do you practice ethical decision-making in your current role?	What is an example?
Evidence	How does evidence influence your decision-making?	As a team? As an organisation?
	How do you currently use data to inform decision-making?	
	Ideally, what data do you need to make the best decisions?	
	How would you define an evidence-based decision for prevention?	What is an example?
	What is an example of a preventive evidence-based decision you have made in your current role?	What data did you use?
Theory (i.e., experiential, and professional judgment)	What is your attitude towards using theory for preventive decision-making?	
	When do your decisions draw upon theory?	
	What is an example of a preventive theory-based decision you have made in your current role?	

**Table 2 ijerph-19-15285-t002:** Participant characteristics in the present study (*n* = 14).

Demographic	*n (%)*
Role category	
Executive	2 (14)
Director/Manager	5 (36)
Project/Clinical Lead	7 (50)
Professional background	
Clinical	4 (29)
Public Health	5 (36)
Information Technology	2 (14)
Research	1 (7)
Project	1 (7)
Operational	1 (7)
Time in current role	
0–6 months	0 (0)
6–12 months	2 (14)
1–2 years	6 (42)
2–5 years	3 (22)
>5 years	3 (22)
Years of experience	
0–2 years	1 (7)
2–5 years	1 (7)
6–10 years	2 (14)
11–20 years	6 (43)
21–30 years	4 (29)
>30 years	0 (0)

**Table 3 ijerph-19-15285-t003:** Themes, sub-themes, and minor theme derived from researcher-led interpretation (stage two) of the Leximancer text analytics.

Theme		Sub-Theme
1—Data	1A	Data is a tool to generate trust in decision-making
	1B	Data storytelling is used to demonstrate the value of prevention
	1C	Better data access and quality is needed to improve the precision of decision-making
	1D	The data paradox: change-agents want more data, but data usability is inversely proportional to data amount
2—Evidence	2A	Traditional evidence is critical to decision-making but is only one piece of a larger evidential puzzle
	2B	Non-traditional sources of evidence (e.g., innovation, experience) are strongly weighted in prevention
Evidence-Ethical (Minor)	EE	The strength of all evidence (traditional and non-traditional) is directly proportional to the ethical rigor of a decision
3—Ethics	3A	Decisions carry a weight of responsibility to the taxpayer, community, and organisation
	3B	Decisions must ultimately be fair and contextualised for equity—to target those with the greatest health need
4—Health	4A	“Health” decisions are driven by everything outside of health
	4B	Healthcare and disease prevention ultimately relates back to dollars

## Data Availability

The data presented in this study are available on request from the corresponding author. The data are not publicly available due to ethics and privacy reasons.

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
