# Peer review of "PREVIDE: A Qualitative Study to Develop a Decision-Making Framework (PREVention decIDE) for Noncommunicable Disease Prevention in Healthcare Organisations"

_ijerph, 2022, doi:10.3390/ijerph192215285_

Round 1

Reviewer 1 Report

- the introduction section contains much-pieced information, i.e, too many short paragraphs. Try to cluster them in logical order.

- is it possible to separate aim 1, i.e., exploration and comparison?

- section 2.2 should describe the 14 participants and convince the audience that 14 is sufficient. 

- how can the framework in figure 2 be applied in other research fields and contexts? 

- it is interesting to see the discussion in 4.2, yet, the number of references seems to be lacking. suggest adding reference on decision-making: 

o) Gray, C. M., & Chivukula, S. S. (2019). Ethical mediation in UX practice. In Proceedings of the 2019 CHI Conference on Human Factors in Computing Systems (pp. 1-11).

o) Ohashi, T., Auernhammer, J., Liu, W., Pan, W., & Leifer, L. (2022). NeuroDesignScience: Systematic Literature Review of Current Research on Design Using Neuroscience Techniques. Design Computing and Cognition’20, 575-592.

Reviewer 2 Report

Thanks for the opportunity for reviewing this study. However, there are some concerns that need to be addressed before publication.

1. Noncommunicable diseases (NCDs) is a general term, please justify how can you generalize it through some frameworks. Alternatively, can you specify which NCDs the authors would like to focus first? eg. Obesity. 2. The sample size (n=14) seems too small for a qualitative study.  It is expected to have at least 18-25 samples. This can refer to these  papers:

1. Chen SC-I, Liu C. Factors Influencing the Application of Connected Health in Remote Areas, Taiwan: A Qualitative Pilot Study. International Journal of Environmental Research and Public Health. 2020; 17(4):1282. https://doi.org/10.3390/ijerph17041282

2. Chen S, Hu R, McAdam R

Smart, Remote, and Targeted Health Care Facilitation Through Connected Health: Qualitative Study

J Med Internet Res 2020;22(4):e14201

URL: https://www.jmir.org/2020/4/e14201

DOI: 10.2196/14201

3. M&M 

Research design: Please justify why the qualitative approach is suitable for this study.

The information of participants are recommended to be illustrated by diagrams.

4. Conclusion: I wonder how such a small sample can draw a valid conclusion. Please justify it.

5. References: some relevant literature can be included. 

qualitative research:

Maxwell, Joseph A. Designing a qualitative study. Vol. 2. The SAGE handbook of applied social research methods, 2008.

Yin, Robert K. Qualitative research from start to finish. Guilford publications, 2015.

Decision making: 

Betsch, Cornelia, et al. "Improving medical decision making and health promotion through culture-sensitive health communication: an agenda for science and practice." Medical Decision Making 36.7 (2016): 811-833.

Preventive medicine:

Tricoli, Antonio, Noushin Nasiri, and Sayan De. "Wearable and miniaturized sensor technologies for personalized and preventive medicine." Advanced Functional Materials 27.15 (2017): 1605271....
